# The Impact of Moderate-Dose Acetylsalicylic Acid in the Reduction of Inflammatory Cytokine and Prevention of Complication in Acute Phase of Kawasaki Disease: The Benefit of Moderate-Dose Acetylsalicylic Acid

**DOI:** 10.3390/children7100185

**Published:** 2020-10-16

**Authors:** Jung Eun Kwon, Da Eun Roh, Yeo Hyang Kim

**Affiliations:** Division of Pediatric Cardiology, Department of Pediatrics, School of Medicine, Kyungpook National University, Kyungpook National University Children’s Hospital, Daegu 41404, Korea; lovecello623@gmail.com (J.E.K.); ponyks1004@naver.com (D.E.R.)

**Keywords:** acetylsalicylic acid, coronary artery, fever, Kawasaki disease

## Abstract

Background: Acetylsalicylic acid (ASA) is part of the recommended treatment of Kawasaki disease (KD). Controversies remain regarding the optimal dose of ASA. We aimed to evaluate the impact of different doses of ASA on inflammation control while minimizing adverse effects in the acute phase treatment of KD. Methods: The enrolled 323 patients with KD were divided into three groups according to ASA dose: moderate-dose (30–50 mg/kg/day), high-dose (80–100 mg/kg/day), and non-ASA. Results: High-dose ASA group showed a significantly shorter duration of fever from the start of treatment to remission than other groups. Baseline level and delta score of interleukin (IL)-1, IL-6, IL-10, tumor necrosis factor-α, and transforming growth factor β were not statistically different among the groups. The number of patients who received additional treatments in the non-ASA group was more than other groups. Coronary artery dilatation was not significantly different among the groups. One patient with high-dose ASA was diagnosed with Reye syndrome. Conclusion: Different doses of ASA did not show any differences in changes of inflammatory bio-makers and cytokines. However, high-dose ASA showed occurrence of Reye syndrome, and non-ASA showed intravenous immunoglobulin refractoriness. We suggest that moderate-dose ASA may be beneficial for the treatment of patients in the acute phase of KD.

## 1. Introduction

Kawasaki disease (KD) is an acute, self-limited systemic vasculitis in children. KD is characterized by fever, erythema of the palms of the hands and soles of the feet, skin rash, bilateral non-exudative conjunctivitis, strawberry tongue, lip redness, and cervical lymphadenopathy.

Vasculitis in KD characteristically occurs in medium-sized arteries, which are primarily the coronary arteries, and can lead to coronary artery dilatation (CAD) or coronary artery aneurysm (CAA). Coronary artery lesions that are complicated with KD are the most common cause of acquired heart disease in children and can cause myocardial infarction or sudden death [1,2]. Intravenous immunoglobulin (IVIG) is the standard treatment for the prevention of coronary artery lesions in the acute phase of KD. Previous studies have reported the incidence of coronary artery lesions to be 20–25% in patients with KD without IVIG treatment and 5% in patients with KD and treated with IVIG [3].

Acetylsalicylic acid (ASA) has been used concomitantly with IVIG to treat patients in the acute phase of KD [1]. The dose of ASA administered varies according to each guideline. High-dose (80–100 mg/kg/day) ASA is widely used in the United States, and moderate-dose (30–50 mg/kg/day) ASA is recommended in Japan [1,4]. High-dose ASA has anti-inflammatory and anti-pyretic effects, which are known to reduce both the duration of fever in the acute phase of KD and the occurrence of coronary artery complications [5]. However, the effect and importance of high-dose ASA in the treatment of the acute phase of KD are controversial. Previous studies have reported complications from high-dose ASA, including an increase of liver enzymes, gastrointestinal bleeding, anemia, and Reye syndrome [3], with no significant reduction in mortality or coronary artery complications when compared to moderate-or low-dose (3–5 mg/kg/day) ASA [2,5,6].

We hypothesized that different ASA doses from high-dose ASA to non-ASA may not show significant differences in decreasing progression because IVIG is the main anti-inflammatory treatment, even though recent KD guidelines have recommended IVIG and ASA combination therapy at the acute phase of KD [1]. We analyzed the changes of biomarker levels, inflammatory cytokines and cardiac complications among patients with different dose of ASA. Based on these results, we aimed to evaluate the impact of different dose of ASA on inflammation control while minimizing adverse effects in the acute phase treatment of KD.

## 2. Results

### 2.1. Patients’ Characteristics

A total of 323 patients were enrolled; 91 patients were in group 1, 97 patients were in group 2, and 135 patients were in group 3. The patients’ general characteristics are described in Table 1. The proportion of male patients was not significantly different among the groups (*p* = 0.378). Mean age of patients was not significantly different among the groups (*p* = 0.127). The number of patients with complete KD was not significantly different among the groups (*p* = 0.683). The mean Kobayashi risk score was not significantly different among the groups (*p* = 0.109), and the number of patients with a Kobayashi risk score ≥4 was not significantly different among the groups (*p* = 0.675).

### 2.2. Fever Duration

The durations of fever, hospitalization, and ASA medication are described in Table 1. The total durations of fever and hospitalization were not significantly different among the groups (*p* = 0.635 and *p* = 0.573, respectively). The time from IVIG administration to the complete subsidence of fever in group 2 was shorter than that in groups 1 and 3 (*p* = 0.004). The duration of ASA administration in groups 1 and 2 was not significantly different (*p* = 0.844).

### 2.3. Control of Inflammatory Biomarkers

#### 2.3.1. Laboratory Data

The baseline laboratory results before IVIG treatment are shown in Table 2. Group 3 had a significantly higher erythrocyte sedimentation rate (ESR), aspartate aminotransferase (AST) and alanine aminotransferase (ALT) level before IVIG treatment than groups 1 and 2 (*p* = 0.001, *p* = 0.003 and *p* = 0.001, respectively). The delta (∆) score for the changes in laboratory values is represented in Table 3. Delta scores for AST and ALT among groups were not significantly different (*p* = 0.050 and *p* = 0.115, respectively).

#### 2.3.2. Relationship between the Change of Cytokine Levels and ASA Dose

The baseline level of each cytokine is described in Table 4. The level of transforming growth factor (TGF)-β1 was significantly different among groups (*p* = 0.046), but the level of the other cytokines was not significantly different among groups. The delta scores of the cytokines showed no significant differences among groups (Table 5).

### 2.4. Complication

#### 2.4.1. Response of IVIG Treatment

Group 3 had a higher number of patients with refractory KD than groups 1 and 2 (*p* = 0.001, Table 6). The main second treatment in groups 1 and 2 was IVIG, and the main second treatment in group 3 was steroids (*p* = 0.001, Table 6).

#### 2.4.2. Cardiac Complications

The number of patients with CAD and CAA was not significantly different among the groups (*p* = 0.688, Table 6).

#### 2.4.3. Occurrence of Reye Syndrome

One case of Reye syndrome occurred in group 2. The patient was a 15-month-old male, and his initial AST/ALT as 23/27 IU/dL. On the 5th day of high-dose ASA administration, the patient showed drowsiness and seizure. Laboratory results showed hyperammonemia and markedly elevated AST/ALT (741/626 IU/dL). The patient was diagnosed with Reye syndrome and died of sepsis after six months of treatment. Groups 1 and 3 had no case of Reye syndrome.

## 3. Discussion

IVIG therapy without ASA when compared to the combination therapy with IVIG and ASA showed the following: (1) no differences in the reduction of C-reactive protein (CRP) levels, inhibition of inflammation as confirmed by cytokine levels, and development of cardiac complications; and (2) a positive effect on the control of elevated liver enzymes. However, IVIG therapy without ASA showed longer fever duration and a higher incidence of additional treatment than the combination therapy with IVIG and ASA. The combination therapy with IVIG and high-dose ASA showed good anti-inflammatory effects but severe fatal complications. The combination therapy with IVIG and moderate-dose ASA showed good anti-inflammatory effects and no significant severe complications.

IVIG is regarded as the standard treatment for acute phase KD to prevent the occurrence of coronary artery lesions [3]. However, high-dose ASA with IVIG has been recommended even though IVIG was established as the primary treatment for acute phase KD [1,5].

ASA can have two effects, which depend on the dose administered. High-dose ASA has potent anti-inflammatory effects and fever control. Low-dose ASA has antiplatelet effects and prevents platelet aggregation at the period of thrombocytosis in the convalescent phase of KD. According to the American Heart Association and European/Japanese guidelines, high-dose (80–100 mg/kg/day) ASA is widely used in the United States, and moderate-dose (30–50 mg/kg/day) ASA is recommended in Japan [1,4]. The Korean Health Insurance Review and Assessment-Pediatric Patients Service of 2010–2015 reported a decreased incidence of non-ASA treatment, and an increased prescription rate of moderate-dose ASA over time in the treatment of acute phase KD in Korea [7]. In the present study, high-dose ASA was mainly used in Center 1; however, since the opening of Center 2 in September 2013, we could observe a gradual increase in the prescriptions of moderate-dose ASA. In addition, an increase in the number of patients treated without ASA in Center 2 starting in 2016 was observed, and this contributed to changes in medical staff and routine practices.

High-dose ASA has adverse effects, such as gastrointestinal bleeding, anemia, and sensorineural hearing loss, and is associated with a higher risk of mortality due to extremely rare but serious complications, such as Reye syndrome [2,8,9,10]. We observed one case of Reye syndrome in the high-dose ASA group. In addition, we previously encountered a child with Reye syndrome that was caused by high-dose ASA who was transferred to a pediatric intensive care unit for treatment; this child was not included in the present study’s data. Although the association with ASA treatment and Reye syndrome in KD is rare and is related to a genetic predisposition and inborn errors of metabolism rather than amount of ASA ingestion, we should make be attentive during ASA treatment in KD.

Several studies have been conducted to assess the different ASA doses used in KD treatment and their associated clinical courses and outcomes [11,12,13,14]. These studies showed no significant difference in the prevalence of CAA or fever duration between high-dose ASA and low-dose ASA in the initial treatment of KD [11,12,13,14]. A previous comparative study between a high-dose ASA group and non-ASA group in the acute phase of KD reported no differences in the reduction of CRP, IVIG responsiveness, or development of CAD; however, the non-ASA group had a longer duration of defervescence than the high-dose ASA [6]. Similar to previous studies, the present study showed no difference in laboratory parameters among the groups after IVIG treatment and no significant differences in the incidence of CAD according to the ASA dose. However, the duration from IVIG administration to fever subsidence was significantly shorter in the ASA group than that in the non-ASA group. We could conclude that high-dose or moderate-dose ASA was an effective anti-pyretic drug in KD.

Abnormal AST/ALT values are commonly found in patients with KD. In the present study, many patients showed varying degrees of abnormal AST/ALT values in all groups, and ESR and AST/ALT were significantly higher in the non-ASA group than ASA groups. We can suspect that the levels of ESR and AST/ALT may be correlated with disease activity in the initial phase. However, CRP as inflammatory indicator and Kobayashi risk score as predictor of IVIG refractoriness were not significantly different among groups. Most patients showed improvement in AST/ALT after initial KD treatment although there was no significant difference, and the non-ASA group had greater improvements in their AST/ALT values than ASA groups. We hypothesized that treatment without ASA could be beneficial in patients with KD and abnormal AST/ALT values.

Several cytokines are involved in the pathophysiology of KD [15,16]. IL-6 and IL-10 have important roles in systemic inflammatory disease, such as pediatric vasculitis. In a recent study, IL-6 markedly increased in the acute phase of KD and decreased after IVIG and high-dose ASA therapy [17]. The present study showed that the levels of IL-6 and IL-10 after IVIG treatment were reduced in all groups and there were no significant differences. These results suggest that ASA may not play a key role in anti-inflammatory effects during the acute phase of KD.

Tumor necrosis factor alpha (TNF-α) is very important in the pathogenesis of KD, especially in cardiac involvement. In a previous Chinese pediatric study of patients with KD, a markedly elevated level of TNF-α was positively correlated with CRP in the acute phase of KD, and an elevated level remained after IVIG therapy; in some patients, TNF-α was related to IVIG-nonresponsive refractory KD and the development of CAD even when other inflammatory cytokines showed no significant relationship to those outcomes [18]. The present study showed that the level of TNF-α in our patients did not significantly change after IVIG treatment, which is similar to the previous study. In addition, the level of TNF-α in our patients was not related to treatment with ASA.

TGF-β is a multifunctional peptide involved in T-cell generation and associated with susceptibility to KD, formation of CAA in KD, aortic root dilatation (Loeys-Dietz syndrome and abdominal aortic aneurysm), and response to IVIG treatment [19,20]. In previous studies, plasma levels of TGF-β1 or 2 were lower in the acute phase of KD than those in the convalescent phase of KD [19,21]. Interleukin (IL)-1 signaling is a new insight in pathogenesis of KD and there was a report that IL-1 inhibition may be related to improvement of pro-inflammatory vasodilatation and myocarditis triggered by IL-1α and IL-1β in a mouse model of KD [22,23,24]. However, in the present study, it was difficult to find statistically significant changes of IL-1 and TGF-β among groups or between before and after IVIG treatment.

The percentage of patients who show resistance to the primary IVIG treatment is approximately 9–30% [25,26,27]. The incidence of IVIG refractoriness in our patients was similar between the high-dose and moderate-dose ASA groups, and similar to previous reports. However, the non-ASA group showed a higher occurrence of IVIG refractoriness than the high-dose and moderate-dose ASA groups. We believe that the high incidence of IVIG refractoriness in the non-ASA group may indicate the need for ASA therapy in patients with KD and may be related to the reduction of treatment without ASA in Korea. Further studies are needed to ascertain the role of early steroid initiation as powerful anti-inflammatory therapy, as well as ASA, to understand the best way of treating KD patients.

### Limitations

The present study has several limitations. First, our study is a retrospective study, thus we lost some detailed information. Second, the number of patients in our study was relatively small; especially, the number of patients with cytokine analysis. Finally, we did not carry out additional studies to confirm the adverse effects of ASA, such as endoscopy or hearing tests.

## 4. Materials and Methods

### 4.1. Study Materials and Data Collection

We conducted a retrospective cohort study of 560 patients treated for either complete or incomplete KD at the Department of Pediatrics, Kyungpook National University Hospital (Center 1) and Kyungpook National University Children’s Hospital (Center 2) from January 2014 to December 2017. One pediatric cardiologist at Center 1 (Doctor A) diagnosed and treated patients from 2014–2016. Two pediatric cardiologists at Center 2 diagnosed and treated patients: Doctor B from 2015–2016 and Doctor C from 2016–2017.

Patients were diagnosed with complete KD when their fever persisted for ≥5 days and at least four of the five typical clinical manifestations based on the 2017 American Heart Association criteria were observed [1]. Patients were diagnosed with incomplete KD when less than four of the five typical clinical manifestations based on the 2017 American Heart Association criteria were observed.

We excluded the following patients: (1) patients who received primary treatment from other hospitals and were transferred to Centers 1 or 2 for secondary treatment, (2) patients who did not undergo blood tests twice in the acute phase of KD (at admission/after IVIG treatment), (3) patients who were admitted during the recovery phase and did not receive IVIG treatment, and (4) patients whose final diagnosis was not KD.

The Kobayashi risk score was calculated for each patient to predict their IVIG resistance using the following parameters: serum sodium, days of illness at initial treatment, AST, percentage of neutrophils, CRP, age, and platelet count [28]. Each parameter has a score of 1 or 2 points, and a cut-off point of ≥4 identifies a patient with a high risk of resistance to IVIG.

The final total of patients enrolled was 323; 97 patients were from Center 1, and 226 patients were from Center 2 (Figure 1).

### 4.2. Treatment

All enrolled patients were treated with IVIG (2 g/kg) as the first-line therapy. During the study period, the treatment policy of the two centers was different and changed according to year. High-dose ASA was frequently used until 2016 in Center 1, and non-ASA use has increased gradually in Center 2. Patients were divided into three groups according to the ASA dose administered in the acute phase of KD. Patients who were treated with moderate-dose (30–50 mg/kg/day) ASA were classified in group 1, patients who were treated with high-dose (80–100 mg/kg/day) ASA were in group 2, and patients who were treated without ASA were in group 3. Patients in groups 1 and 2 were administered their corresponding dose of ASA up to 48 h after their fever subsided from the day of IVIG treatment. After their fever subsided, all patients were prescribed low-dose (3–5 mg/kg/day) ASA that was administered as a single dose per day for 6–8 weeks.

Refractory KD was considered if persistent or recrudescent fever was noted at least 36 h after initial IVIG administration [1]. These patients were treated with second-line treatment such as additional IVIG infusions, systemic steroids or infliximab.

### 4.3. Laboratory Tests

We performed a blood test on the first day of admission prior to the administration of IVIG, and a follow-up blood test was performed at least 24 h after the administration of IVIG. We evaluated the total white blood cell counts (WBC), proportions of neutrophils and lymphocytes, hemoglobin levels, platelet counts, erythrocyte sedimentation rate, CRP, AST, and ALT. The change in each laboratory value was represented by the delta (∆) score.

The normal ranges of AST and ALT vary according to a patient’s age. The normal ranges of AST were <63 IU/dL for patients aged 1–12 months, <60 IU/dL for patients aged 1–3 years, <50 IU/dL for patients aged 3–9 years, and <40 IU/dL for patients aged 9–15 years. The normal range of ALT was <45 IU/dL for all patient age groups [29].

We investigated the cytokine levels in 67 patients: 20 patients in group 1, 8 patients in group 2, and 39 patients in group 3. The serum that remained after the above laboratory tests were performed was stored at −80 °C until further analysis. Serum levels of IL-1a, IL-1b, IL-6, IL-10, TNF-α, TGF-β 1, TGF- β 2, and TGF-β 3 were measured using a multiplex assay on the Luminex^®^ 200™ Total System (Luminex Corporation, Austin, TX, USA) according to the manufacturer’s instructions.

### 4.4. Echocardiography

Cardiac and coronary artery complications were evaluated by echocardiography during hospitalization. We used the coronary artery diameter and aortic valve annular diameter to determine the z-score for confirmation of CAD [30]. We obtained the luminal diameters of the proximal right coronary artery, left main coronary artery, proximal left anterior descending artery, and proximal left circumflex artery. The aortic valve annular diameters were measured with magnification in parasternal long-axis views from the inner edge of the proximal valve insertion hinge point within the arterial root to the inner edge of the opposite hinge point. CAD was confirmed when the z-score of the coronary artery size was >2.0. CAA was defined as a z-score of the coronary artery size >2.5 [1].

### 4.5. Primary and Secondary Outcomes of the Study

The primary outcome was to assess that all kinds of ASA dose can sufficiently control inflammation without lengthening fever duration. The secondary outcome was to confirm that there is no difference of the incidence of IVIG resistance or cardiac complications according to ASA dose.

### 4.6. Statistical Analysis

All statistical analyses were performed using SPSS version 23.0 (SPSS for Windows, version 23.0, SPSS Inc., Chicago, IL, USA). The results are presented using means and standard deviations. The one-way ANOVA test was used for the comparisons of patient characteristics, and the comparative analysis was performed using the Student’s *t*-test and Linear-by-Linear association method. The changes of laboratory test values before and after IVIG treatment were compared using the paired *t*-test and Wilcoxon signed rank test. A *p*-value of <0.05 was considered statistically significant.

### 4.7. Ethics Statement

This study protocol was reviewed and approved by the Institutional Review Board of Kyungpook National University Hospital (approval No. KNUH 2018-01-035) and Kyungpook National University Chilgok Hospital (approval No. KNUCH 2018-01-032).

## 5. Conclusions

In summary, this study demonstrated that different doses of acetylsalicylic acid did not show any differences in changes of inflammatory biomarker and cytokines, although previously published guidelines recommend ASA as a primary treatment of KD. Before treatment with ASA, we must consider the following: high-dose ASA may result in Reye syndrome and sometimes should be avoided to prevent severe complications, especially when influenza virus infection or chickenpox is prevalent regionally or seasonally; and treatment with non- ASA may be helpful in patients with KD and acute hepatitis, but can result in prolonged fever and frequent IVIG refractoriness. We suggest that treatment with ASA should depend on the patient’s clinical situation rather than serve as a universal application for all patients with KD; in addition, moderate-dose ASA may be more beneficial for the treatment of patients in the acute phase of KD than high-dose or non-ASA treatment.

## Figures and Tables

**Figure 1 children-07-00185-f001:**
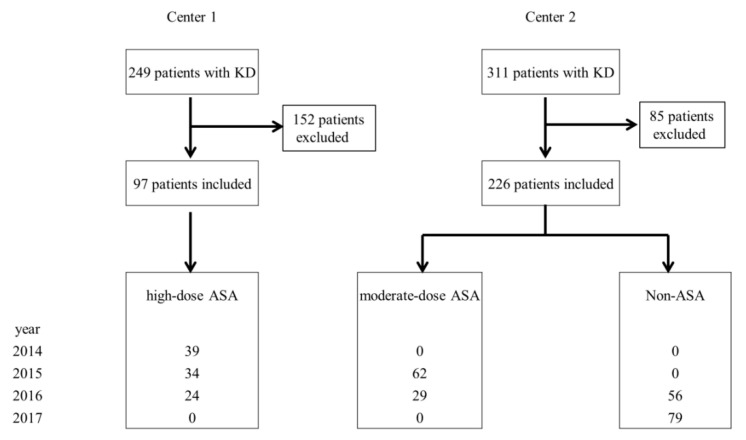
Subject disposition chart. Values of the lowest box are presented as year and number of patients. KD, Kawasaki disease; ASA, acetylsalicylic acid.

**Table 1 children-07-00185-t001:** General characteristics of Kawasaki disease patients by group.

	Group 1 (n = 91)	Group 2 (n = 97)	Group 3 (n = 135)	*p* Value
Mean age at diagnosis (months)	34.2 ± 30.8	32.8 ± 22.9	39.6 ± 30.4	0.127
Male (n)	54 (59.3)	66 (68.0)	75 (55.6)	0.378
Complete KD (n)	62 (68.1)	71(73.2)	97 (71.9)	0.683
Fever				
Total duration of fever (day)	5.9 ± 2.0	5.9 ± 2.6	6.1 ± 2.5	0.635
Duration from IVIG administration to fever subside (hours)	37.7 ± 47.3	26.8 ± 35.7	46.6 ± 47.9	0.004 *
Kobayashi risk score	3.0 ± 2.0	2.4 ± 2.1	2.7 ± 2.0	0.145
≥4 Kobayashi risk score (n)	33 (36.3)	31 (32.0)	43 (31.9)	0.569
Hospitalization duration (day)	6.2 ± 6.7	5.9 ± 5.6	5.5 ± 2.7	0.573
Duration of ASA medication (day)	3.7 ± 1.7	3.7 ± 2.0	−	0.844

Values are presented as mean ± standard deviation or number (%). Group 1; patients treated with moderate-dose (30–50 mg/kg/day) ASA, Group 2; patients who treated with high-dose (80–100 mg/kg/day) ASA, Group 3; patients who treated without ASA. n = number, KD = Kawasaki disease, IVIG = intravenous immunoglobulin, ASA = Acetylsalicylic acid. * *p* < 0.05 among groups.

**Table 2 children-07-00185-t002:** Baseline laboratory data before intravenous immunoglobulin (IVIG) treatment.

	Group 1 (n = 91)	Group 2 (n = 97)	Group 3 (n = 135)	*p* Value
WBC (/mm^3^)	14,106 ± 5,469	13,724 ± 5,139	14,243 ± 6,016	0.782
Neutrophil (%)	63.6 ± 16.8	64.5 ± 16.2	66.3 ± 16.9	0.481
Hemoglobin (g/dL)	11.5 ± 0.2	11.8 ± 1.0	12.5 ± 8.6	0.387
Platelet (×10^3^/mm^3^)	356 ± 110	382 ± 127	365 ± 121	0.324
ESR (mm/hr)	46.8 ± 28.7	37.9 ± 19.9	53.5 ± 28.5	0.001 *
CRP (g/dL)	6.9 ± 5.7	6.3 ± 7.8	8.2 ± 7.2	0.108
AST (IU/L)	70.5 ± 97.4	61.5 ± 81.4	136.9 ± 264.2	0.003 *
ALT (IU/L)	66.0 ± 99.0	70.0 ± 110.3	133.3 ± 195.6	0.001 *

Values are presented as mean ± standard deviation. Group 1; patients treated with moderate-dose (30–50 mg/kg/day) ASA, Group 2; patients treated with high-dose (80–100 mg/kg/day) ASA, Group 3; patients treated without ASA. IVIG = intravenous immunoglobulin, WBC = white blood cell, ESR = erythrocyte sedimentation rate, CRP = C-reactive protein, AST = aspartate aminotransferase, ALT = alanine aminotransferase. * *p* < 0.05 among groups.

**Table 3 children-07-00185-t003:** Comparison laboratory changes among groups.

	Group 1 (n = 91)	Group 2 (n = 97)	Group 3 (n = 135)	*p* Value
∆WBC (/mm^3^)	−5308 ± 4958	−5483 ± 4850	−5110 ± 5609	0.870
∆Neutrophil (%)	−26.2 ± 19.3	−26.3 ± 18.8	−23.6 ± 18.2	0.467
∆Hemoglobin (g/dL)	−0.4 ± 1.0	−0.4 ± 0.9	−1.3 ± 8.5	0.362
∆Platelet (×10^3^/mm^3^)	22.2 ± 103.5	31.0 ± 95.5	32.8 ± 102.8	0.718
∆ESR (mm/hr)	8.1 ± 21.7	10.3 ± 28.0	19.0 ± 27.5	0.005 *
∆CRP (g/dL)	−3.0 ± 7.3	−3.3 ± 6.7	−4.5 ± 4.8	0.155
∆AST (IU/L)	−36.1 ± 88.8	−42.5 ± 111.5	−97.2 ± 252.8	0.050
∆ALT (IU/L)	−43.6 ± 97.5	−36.1 ± 84.6	−69.5 ± 136.1	0.115

Values are presented as mean ± standard deviation. Group 1; patients treated with moderate-dose (30–50 mg/kg/day) ASA, Group 2; patients treated with high-dose (80–100 mg/kg/day) ASA, Group 3; patients treated without ASA. WBC = white blood cell, ESR = erythrocyte sedimentation rate, CRP = C-reactive protein, AST = aspartate aminotransferase, ALT = alanine aminotransferase. * *p* < 0.05 among groups.

**Table 4 children-07-00185-t004:** Baseline cytokine level before IVIG treatment.

	Group 1 (n = 20)	Group 2 (n = 8)	Group 3 (n = 39)	*p* Value
IL-1a (pg/mL)	14.6 ± 26.8	3.0 ± 4.4	15.7 ± 43.2	0.666
IL-1b (pg/mL)	2.5 ± 2.9	0.9 ± 1.2	10.3 ± 50.0	0.684
IL-6 (pg/mL)	56.8 ± 45.8	100.5 ± 113.6	98.4 ± 113.5	0.285
IL-10 (pg/mL)	96.6 ± 116.9	81.2 ± 62.0	138.4 ± 207.2	0.551
TNF-α (pg/mL)	25.6 ± 9.9	19.2 ± 10.9	45.8 ± 83.0	0.380
TGF-β1 (pg/mL)	47,396 ± 10,033	34,821 ± 11,827	40,892 ± 13,985	0.046 *
TGF-β2 (pg/mL)	2619 ± 1050	1666 ± 707	2442 ± 934	0.059
TGF-β3 (pg/mL)	57.2 ± 27.7	44.3 ± 12.7	47.9 ± 15.6	0.160

Values are presented as mean ± standard deviation. Group 1; patients treated with moderate-dose (30–50 mg/kg/day) ASA, Group 2; patients treated with high-dose (80–100 mg/kg/day) ASA, Group 3; patients treated without ASA. IVIG = intravenous immunoglobulin, IL = interleukin, TNF = tumor necrosis factor, TGF = transforming growth factor. * *p* < 0.05 among groups.

**Table 5 children-07-00185-t005:** Comparison changes of cytokine level among groups.

	Group 1 (n = 20)	Group 2 (n = 8)	Group 3 (n = 39)	*p* value
∆IL-1a (pg/mL)	7.0 ± 39.7	15.6 ± 38.5	9.9 ± 56.1	0.920
∆IL-1b (pg/mL)	1.0 ± 7.0	−0.2 ± 0.8	−6.4 ± 49.1	0.750
∆IL-6 (pg/mL)	−48.6 ± 46.7 *	−94.1 ± 114.0	−84.2 ± 110.4	0.345
∆IL-10 (pg/mL)	−75.0 ± 116.3	−65.2 ± 57.4	−107.7 ± 214.1	0.718
∆TNF-α (pg/mL)	7.1 ± 19.3	1.7 ± 14.5	−12.8 ± 77.2	0.468
∆TGF-β1 (pg/mL)	−5450 ± 7187	1911 ± 11,928	−1683 ± 13,932	0.304
∆TGF-β2 (pg/mL)	−619.6 ± 530.9	−192.6 ± 598.5	−435.7 ± 718.6	0.284
∆TGF-β3 (pg/mL)	8.1 ± 31.0	14.0 ± 55.4	18.2 ± 21.7	0.478

Values are presented as mean ± standard deviation. Group 1; patients treated with moderate-dose (30–50 mg/kg/day) ASA, Group 2; patients treated with high-dose (80–100 mg/kg/day) ASA, Group 3; patients treated without ASA. IL = interleukin, TNF = tumor necrosis factor, TGF = transforming growth factor. * *p* < 0.05 among groups.

**Table 6 children-07-00185-t006:** Comparison of clinical prognosis among groups.

	Group 1 (n = 91)	Group 2 (n = 97)	Group 3 (n = 135)	*p* Value
IVIG resistant	16 (17.6)	20 (20.6)	67 (49.6)	0.001 *
Second treatment				
IVIG	16 (17.6)	20 (20.6)	16 (11.9)	
Steroid	0 (0)	0 (0)	51 (37.8)	0.001 *
Coronary artery dilatation	7 (8.0)	5 (5.1)	7 (5.2)	0.688
2.0 < z score ≤ 2.5	2 (2.5)	1 (1.0)	1 (0.7)	
2.5 < z score	5 (5.5)	4 (4.1)	6 (4.4)	

Values are presented as number (%). Group 1; patients treated with moderate-dose (30–50 mg/kg/day) ASA, Group 2; patients treated with high-dose (80–100 mg/kg/day) ASA, Group 3; patients treated without ASA. n = number, IVIG = intravenous immunoglobulin. * *p* < 0.05 among groups.

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
