# Peer review of "The Impact of Moderate-Dose Acetylsalicylic Acid in the Reduction of Inflammatory Cytokine and Prevention of Complication in Acute Phase of Kawasaki Disease: The Benefit of Moderate-Dose Acetylsalicylic Acid"

_children, 2020, doi:10.3390/children7100185_

Round 1

Reviewer 1 Report

Would suggest to stress the differences in ESR and liver enzymes in patients not treated with aspirin this may indicate that patients without ASA were sicker.

Attributing fatality to Reye when the fatality occured 6 months later should be reexamined, probably Reye was the cause but I am not sure you can attribute mortality to aspirin.

Author Response

Ms. Jolin Tian

Assistant Editor of Children

Dear Ms. Jolin Tian,                                                                     Oct 07. 2020

Thank you for the review of our manuscript children-945521, titled “The Impact of Moderate-Dose Acetylsalicylic Acid in the Reduction of Inflammatory Cytokine and Prevention of Complication in Acute Phase of Kawasaki Disease: The Benefit of Moderate-Dose Acetylsalicylic Acid”. We have performed the corrections/modifications in response to the comments raised by the Editor and the Reviewers. The point-by-point responses with these comments are indicated hereafter in blue. Changes are yellow-highlighted through the revised manuscript.

*******************************************************************************************************************

Reviewer #1

  1. Would suggest to stress the differences in ESR and liver enzymes in patients not treated with aspirin this may indicate that patients without ASA were sicker.

>> We appreciate and agree with the reviewer’s suggestion. We already described those results in the Results section (Line 82-84) as “Group 3 had a significantly higher ESR, AST and ALT level before IVIG treatment than groups 1 and 2 (P = 0.001, P = 0.003 and P = 0.001, respectively).” However, CRP as inflammatory indicator (Table 2, P = 0.108) and Kobayashi risk score as predictor of IVIG refractoriness (Line 65-68, P = 0.109) were not significantly different between groups. We can suspect that the levels of ESR and liver enzymes may be correlated with disease activity of initial phase although we do not know relation between the levels of ESR/liver enzymes and IVIG refractoriness.

Accordingly, we have described the details in the Discussion section (Line 178-185) as “In the present study, many patients showed varying degrees of abnormal AST/ALT values in all groups, and ESR and AST/ALT were significantly higher in the non-ASA group than ASA groups. We can suspect that the levels of ESR and AST/ALT may be correlated with disease activity of initial phase. However, CRP as inflammatory indicator and Kobayashi risk score as predictor of IVIG refractoriness were not significantly different between groups. Most patients showed improvement of AST/ALT after initial KD treatment although there was no significant difference, and the non-ASA group had greater improvements in their AST/ALT values than the ASA group.”

  1. Attributing fatality to Reye when the fatality occurred 6 months later should be reexamined, probably Reye was the cause but I am not sure you can attribute mortality to aspirin.

>> Thank you very much for the reviewer’s comment. We agree to your opinion about occurrence of fatality to aspirin. Actually, there was adverse drug reaction rather than life-threatening event. We revised the term ‘Life-threatening complications’ to ‘Occurrence of Reye syndrome’ in the Abstract (Line 25-26) and the Results 2.4.3 section (Line 126 and 131).

*******************************************************************************************************************

The authors want to extend our sincere thanks to the Editor and the Reviewers for the helpful comments and the time they invested in reviewing our paper. We hope that this revision will satisfy the comments and requests from the Editor and the Reviewers as well as improve the overall quality of this manuscript. We also hope that the revised manuscript is now acceptable for publication in Children.

Sincerely yours,

Yeo Hyang Kim, MD.,Ph.D.

Department of Pediatrics,

School of Medicine, Kyungpook National University,

Division of Pediatric Cardiology,

Kyungpook National University Children’s hospital, Daegu, Republic of Korea

680 Gukchaebosang-ro, Jung-gu,

Daegu, 41944, Korea

Tel: 82-53-200-2747

Fax: 82-53-425-6683

Reviewer 2 Report

Well written and well conducted study. The different dosages of aspirin therapy in Kawasaki disease had long raised
perplexity.
This study scientifically demonstrates what clinicians, at least in Europe,
had long ago empirically suspected and then modified in their daily practice. Review in the caption of the tables "Group 2; patients who treated with moderate-dose (80-100 mg / kg / d)"
instead of high dose-

Finally, the fact of having a control group 3 without aspirin therapy,
with more frequent persistence of fever, as in the forms of resistant KD,
underlines the importance of a powerful anti-inflammatory therapy:
IVIG + aspirin versus IVIG + steroids.
Furher studies will help us to understand the best way of
treating KD patients

Author Response

Ms. Jolin Tian

Assistant Editor of Children

Dear Ms. Jolin Tian,                                                     Oct 07. 2020

Thank you for the review of our manuscript children-945521, titled “The Impact of Moderate-Dose Acetylsalicylic Acid in the Reduction of  Inflammatory Cytokine and Prevention of Complication in Acute Phase of Kawasaki Disease: The Benefit of Moderate-Dose Acetylsalicylic Acid”. We have performed the corrections/modifications in response to the comments raised by the Editor and the Reviewers. The point-by-point responses with these comments are indicated hereafter in blue. Changes are yellow-highlighted through the revised manuscript.

*******************************************************************************************************************

Reviewer #2

Well written and well conducted study.

The different dosages of aspirin therapy in Kawasaki disease had long raised perplexity.
This study scientifically demonstrates what clinicians, at least in Europe, had long ago empirically suspected and then modified in their daily practice.

  1. Review in the caption of the tables "Group 2; patients who treated with moderate-dose (80-100 mg / kg / d)" instead of high dose-

>> We apologize for this error and have corrected the indicated sentence. Moderate-dose (80-100 mg / kg / d) in each table was revised to high dose (80-100 mg / kg / d).

  1. Finally, the fact of having a control group 3 without aspirin therapy, with more frequent persistence of fever, as in the forms of resistant KD, underlines the importance of a powerful anti-inflammatory therapy: IVIG + aspirin versus IVIG + steroids
    Further studies will help us to understand the best way of treating KD patients

>> We appreciate and agree with the reviewer’s point. We are working on another study about Kawasaki disease and IVIG + aspirin versus IVIG + steroids. If we successfully finish the study, we will submit these findings as a new article on ‘Children’ or MPDI’s journal. We also mentioned that further studies are needed to ascertain the role of early steroid for anti-inflammatory therapy as well as ASA in the Discussion section (Lines 216-218).

*******************************************************************************************************************

The authors want to extend our sincere thanks to the Editor and the Reviewers for the helpful comments and the time they invested in reviewing our paper. We hope that this revision will satisfy the comments and requests from the Editor and the Reviewers as well as improve the overall quality of this manuscript. We also hope that the revised manuscript is now acceptable for publication in Children.

Sincerely yours,

Yeo Hyang Kim, MD.,Ph.D.

Department of Pediatrics,

School of Medicine, Kyungpook National University,

Division of Pediatric Cardiology,

Kyungpook National University Children’s hospital, Daegu, Republic of Korea

680 Gukchaebosang-ro, Jung-gu,

Daegu, 41944, Korea

Tel: 82-53-200-2747

Fax: 82-53-425-6683
